# cAMP-Dependent Signaling and Ovarian Cancer

**DOI:** 10.3390/cells11233835

**Published:** 2022-11-29

**Authors:** Agnieszka Kilanowska, Agnieszka Ziółkowska, Piotr Stasiak, Magdalena Gibas-Dorna

**Affiliations:** Department of Anatomy and Histology, Collegium Medicum, University of Zielona Gora, 65-046 Zielona Gora, Poland

**Keywords:** ovarian cancer, cAMP, PKA, CREB, EPAC

## Abstract

cAMP-dependent pathway is one of the most significant signaling cascades in healthy and neoplastic ovarian cells. Working through its major effector proteins—PKA and EPAC—it regulates gene expression and many cellular functions. PKA promotes the phosphorylation of cAMP response element-binding protein (CREB) which mediates gene transcription, cell migration, mitochondrial homeostasis, cell proliferation, and death. EPAC, on the other hand, is involved in cell adhesion, binding, differentiation, and interaction between cell junctions. Ovarian cancer growth and metabolism largely depend on changes in the signal processing of the cAMP-PKA-CREB axis, often associated with neoplastic transformation, metastasis, proliferation, and inhibition of apoptosis. In addition, the intracellular level of cAMP also determines the course of other pathways including AKT, ERK, MAPK, and mTOR, that are hypo- or hyperactivated among patients with ovarian neoplasm. With this review, we summarize the current findings on cAMP signaling in the ovary and its association with carcinogenesis, multiplication, metastasis, and survival of cancer cells. Additionally, we indicate that targeting particular stages of cAMP-dependent processes might provide promising therapeutic opportunities for the effective management of patients with ovarian cancer.

## 1. Introduction

As provided by Global Cancer Statistics in 2020, ovarian cancer (OC) is the 8th most commonly occurring cancer with the mortality rate accounting for 4.7% of the entire cancer-related deaths among females [1]. In most cases of OC, the prognosis is poor, which results directly from late diagnosis, i.e., in the third or fourth stage of the disease. Conventional therapy for advanced OC is based on cytoreductive surgery combined with platinum-based chemotherapy, however, approximately 50% of patients appear to be chemoresistant and 60% of patients must face cancer recurrence [2]. Much research has focused on the potential causes of ovarian cancer, highlighting the role of key risk factors including family history, ethnicity, and hereditary conditions, particularly mutations in BRCA1/2 (breast cancer 1/2) and MMR (mismatch repair) genes, as well as age, obesity, endometriosis, diet, anti-inflammatory drugs, postmenopausal hormone therapy, smoking, chronic inflammation and infections in the ovary, including HPV, CMV, and *chlamydia trachomatis*. Also, a greater tendency to develop ovarian cancer has been noted in nulliparous women as compared with parous women and in women being on ovarian stimulating drugs used for infertility treatment [3,4,5]. Interestingly, a significant reduction in the risk of developing an epithelial type of ovarian cancer (EOC) has been demonstrated in women using oral contraceptives for more than 10 years [6,7].

Most ovarian tumors arise from tissues in their immediate topographic vicinity, i.e., the fallopian tubes, the endometrium, or the junction of the fallopian tubes with the mesothelium. Initially (in the late 1980s and early 2000s) it has been suggested that approximately 90% of EOC develop from the ovarian surface epithelium (OSE). Further research confirmed however that high-grade serous ovarian carcinomas originate in the fallopian tube [7,8,9]. The tissue of origin became the basis for traditional cancer classification determining a histological type of a tumor. That is why ovarian cancer is characterized by such a high heterogeneity [10] and different level of tumorigenic differentiation associated with precursor changes, pathogenesis, mode of spread, sensitivity to chemotherapy, and further prognosis [11,12].

In 2020, the 5th edition of the World Health Organization Classification of Tumors established a coherent cancer classification, including female genital tumors, in which five main histotypes of ovarian cancer were distinguished: high-grade serous cancer (HGSC), low-grade serous cancer (LGSC), mucous cancer (MC), endometrioid carcinoma (EC) and clear cell carcinoma (CCC). This classification is based on the recognition of specific phenotypes and is the basis for distinguishing ovarian cancer subclasses, i.e., molecular subtypes. For each of them, the source of origin was described and immunohistochemical markers (WT1/p53/napsin A/PR) were recognized allowing for reliable identification of a specific histotype with high accuracy [13]. Cancer development is influenced by many factors including genomic mutations and epigenetic changes which are the features that enable the identification of subtypes [13]. The research focused on mutations as epigenetic changes may provide detailed information relevant to early diagnosis and treatment response [14].

For these reasons, the elucidation of the mechanisms driving ovarian cancer development and growth is particularly important to establish effective targeted treatment [15]. Signal transduction pathways (STPs) control and participate in many cellular processes, including cell division and differentiation, migration, and metabolism. Hypo- or hyperactivation of certain STPs may be associated with uncontrolled tumor growth and metastatic spread. Therefore, it is believed that STPs may serve as good targets for an effective treatment strategies. The cAMP-dependent pathway (cAMP/PKA/CREB pathway) appears to have tremendous therapeutic potential, especially in terms of targeted anticancer therapy. In the neoplastic cells, cAMP signaling modulates and promotes growth, proliferation, metastasis, and invasion but most importantly, it may suppress apoptosis, creating favorable conditions for ovarian tumor progression [16]. Moreover, high expression and activation of cAMP downstream effectors, particularly CREB, are associated with the development of the most aggressive type of tumor and decreased survival chances [17,18,19]. In addition, it has been shown that CREB may suppress apoptosis, leading to an increased rate of relapse and poor survival of OC patients. Conversely, selective phosphodiesterase inhibitors targeting the PDE4 isoform, display apoptotic effects and inhibit proliferation according to the up-regulated expression of mitochondrial ferritin via cAMP/PKA/CREB signals [15]. The cAMP pathway and its downstream effectors may either inhibit or promote cancer [20] and detailed molecular mechanisms that may depend on tumor localization and type, or downstream target genes, have not been elucidated.

## 2. cAMP/PKA/CREB Cascade Signaling

Growth factors, neurotransmitters, hormones, and ions may act as first messengers that initiate cell signaling via triggering a series of signal transduction cascades through the membrane or intracellular receptors. Activated receptors initiate further effects, including the formation of the second messenger molecules such as cAMP, cGMP, inositol triphosphate (IP3), diacylglycerol (DAG), and stimulation of the second messenger system [16] that operates in response to the entry of calcium ions into the cells [20]. cAMP signaling cascade is one of the most essential and best-understood cellular pathways which mediate the action of a wide variety of hormones, neurotransmitters, and growth factors [21,22]. Moreover, the presence of cAMP is essential for the control and regulation of gene expression, growth, cell differentiation as well as proliferation and apoptosis [23,24,25]. The versatility of cAMP-related effects depends on the expression of many factors: adenylyl cyclase isoform [26], phosphodiesterase [27], and A kinase-anchoring proteins (AKAP) [28]. Two families of adenylyl cyclase contribute to the production of cAMP, transmembrane adenylyl cyclases (tmACs, ADCY1-9), and, localized in several intracellular compartments, including cytosol, nucleus, and mitochondria, soluble adenylyl cyclase (sAC, ADCY10) [29,30,31,32]. Various isoforms of adenylate cyclase (AC) are located in the plasma membrane microdomains that along with specific GPCRs shape the signaling complexes involved in the cAMP signaling compartments [33]. Such a system guarantees the diversity of cellular and physiological responses depending on different cAMP pools. When the membrane G protein-coupled receptors (GPCRs) are activated by external stimuli, eg. hormones or neurotransmitters, their heterotrimeric G protein alpha subunits (Gα) stimulate tmAC leading to downstream signaling by cAMP [34]. The structure of the main tmAC domains comprises the intracellular N-terminus and the coiled-coil helical domain, located between the two clusters of six transmembrane helices derived from the C1 and C2 cytoplasmic loops [32,35]. In contrast, soluble adenylyl cyclase is not sensitive to G protein regulation but is directly activated by HCO3- and Ca^2+^ ions at various intracellular locations [27,28]. Although sAC activity is also associated with cAMP-dependent signal transduction, the two types of adenylyl cyclase work independently [34,36]. Studies have shown that sAC working through CFTR/HCO3-/sAC cascade strongly affects FSH signaling, and a number of phosphorylation events are associated with cAMP/PKA activity in the ovary. It has been shown that cytoplasmic cAMP, which is generated by tmAC, activates the transmembrane conductance regulator (CFTR) on the plasma membrane which allows HCO3- ions influx. HCO3- easily enter the nucleus amplifying the activity of nuclear cAMP/PKA/CREB and cAMP/PKA/ERK pathways leading to enhanced granulosa cell proliferation and estrogen synthesis [37]. The sAC-cAMP-PKA signaling is also present within the mitochondria and is activated in response to bicarbonate ions, or more precisely, to metabolically produced COLocally generated mitochondrial cAMP significantly modulates oxidative phosphorylation adapting its rate to the substrate availability, thus mito-sAC may control reactive oxygen species production and contribute to the energy metabolism of the cell [30]. However, since tmAC is the most widely studied source of cAMP, the majority of described cAMP-dependent cellular responses are due to this type of AC activity, whereas biochemical regulation as well as the physiological function of sAC require further investigation [32,36]. 

PDE, being responsible for the hydrolysis of cAMP to its inactive 5’-monophosphate, is a potent factor limiting the intracellular concentration of cAMP [38,39,40]. PDEs are a large group of enzymes made up of 11 families (i.e., PDE1–PDE11) divided into subgroups (containing isoforms numbered 1–4) encoded by different genes. Tissue distribution and substrate specificity (cAMP and/or cGMP) are recognized as characteristic features of the individual family of enzymes. PDE4, PDE7 and PDE8 hydrolyze only cAMP; cGMP is the substrate for PDE5, PDE6 and PDE9, while PDE1, PDE2, PDE3, PDE10 and PDE11 are the two-substrate enzymes. In addition, PDE7 and PDE8 may work together and regulate cAMP levels. In the ovary, PDEs act as indispensable regulators of follicular growth, maturation, and ovulation [41,42]. A high expression of ovarian PDE8A and PDE8B (approximately 77.0% of all PDE mRNAs) and quite a high expression of PDE4D isoforms were confirmed by quantitative PCR (qPCR). Petersen et al. [43] observed that the activity of PDE4 is higher at high intracellular concentrations of cAMP whereas PDE7/8 operates better at lower cAMP levels. Distribution of PDE isoenzymes in the ovary varies depending on their family and subtype. For example, PDE3B, PDE8A, and PDE8B were found in theca cells and oocytes, PDE4A in medullar stromal tissue and oocytes, PDE4B in theca cell layer, PDE4C, and PDE4D in ovarian follicles, while PDE7A and PDE7B in oocytes [43]. 

In the ovary, the two main types of cAMP effector proteins have been recognized - cAMP-dependent protein kinase A (PKA) and exchange protein directly activated by cAMP (EPAC). PKA, as a tetrameric holoenzyme, is composed of two regulatory (R) subunits in a homodimer shape (RIα/RIβ in PKA type I, and RIIα/RIIβ in PKA type II) and two catalytic (C) subunits which are capable of phosphorylating serine/threonine located in the target protein [44]. The binding of cAMP to the corresponding domains of the R subunit unblocks the active site on the C subunit by a conformational change which is manifested by allosteric displacement of the inhibitory sequence and, consequently, induces activation of PKA [25]. Typically, these domains are located around kinase A-anchoring protein (AKAP) which serves as a scaffold [44]. AKAPs play an essential role in signal compartmentalization. They are responsible for the binding of PKA holoenzyme to specific subcellular sites. This function is essential for PKA as it increases signaling specificity and efficiency. The interaction between AKAP and PKA occurs through the binding of AKAP amphipathic α-helix to N-terminal Dimerization/Docking (D/D) domains of PKA regulatory (R) subunits. AKAP affects subcellular PKA activity levels through the spatial targeting of PKA, extending the site of interaction to various cellular locations, such as plasma membrane, mitochondria, endo/sarcoplasmic reticulum, or nucleus [45,46]. That is, to the target site where the locally specific cAMP signaling takes place [47,48]. Another cAMP effector is the exchange protein activated by cAMP (EPAC), also known as the protein cAMP-regulated guanine nucleotide exchange factor (cAMP-GEF) [49]. Its function is related to GDP/GTP exchange and further activation of Ras family proteins, small GTPases called Rap1 and Rap2 [50,51]. It has been shown that EPAC activates Rap1 independently of PKA, causing the phosphorylation of GDP. PKA activation, by inhibiting GDP phosphorylation, exerts the opposite effect. Two isoforms of human EPAC proteins have been described. EPAC1 with a single cAMP binding domain and EPAC2 containing two cAMP binding domains [52]. Apart from the cAMP binding domain at the NH2-terminus, EPAC contains an enzymatic GEF domain at the COOH-terminus for the small Rap1 protein. The binding of cAMP to EPAC causes it to assume the open conformation, which enables Rap binding and activation. In contrast, cAMP deficiency results in a closed conformation of the cAMP binding domain, which blocks Rap binding [53,54]. The activity of the cAMP binding domain induces a serious conformational change, and its stabilization results from the interaction with the adenosine group [55]. Studies have shown a different expression pattern for EPAC1 and EPAC2 in mature and developing cells. In addition, each isoform represents different expressions depending on tissue localization in the body, i.e.: EPAC1 mRNA is present in all tissues [56], whereas EPAC2 mRNA is mainly in the brain and endocrine tissues [57]. Also, research has revealed diverse locations of EPAC1 and EPAC2 in many subcellular regions, including the plasma membrane, cytosol, mitochondria, and nuclear membrane [58]. The location of EPAC1 has been identified in a specific intracellular domain where EPAC1 enables binding to scaffold proteins and phosphodiesterase 4D [59]. EPAC1, as a cAMP-dependent protein, is controlled by cAMP for its activity and translocation from the cytosol to the plasma membrane [60]. Several membrane anchors have been described for the EPAC1 protein, suggesting its different localization depending on the conditions [61].

cAMP-related cellular functions of EPAC include the facilitation of cell adhesion, modulation of cell proliferation and differentiation, and interactions between cell junctions. Similarly, to PKA, EPAC also affects cyclic-AMP response element binding (CREB) protein, which is active in many cellular processes including gene transcription, cell migration, mitochondrial homeostasis, cell proliferation, and death. CREB along with c-Myc, c-Fos, and c-Jun, belong to the basic leucine zipper (bZIP) transcription factors closely cooperating with CREM (cAMP response element modulator) and ATF-1 (activating transcription factor 1) [62]. These factors are characterized by the presence of the C-terminal basic domain mediating DNA binding, as well as the leucine zipper domain, which facilitates dimerization [63] CREB modular protein comprises kinase-induced domain (KID), two glutamine-rich domains, and a bZIP domain. Domains containing KID and glutamine are necessary for CREB transactivation and phosphorylation. Reversible CREB phosphorylation by e.g., PKA, AKT, MAPK, and the 90 kD ribosomal S6 kinase of various serine residues induces the transcriptional activity of CREB [64,65]. CREB-dependent initiation of gene transcription is stimulated by CREB/CBP complex which recruits transcription machinery in gene promoters, specifically at the CRE site [66]. As a major activator of transcription, CREB is involved in the modulation of histone H3 and H4 methylation, which is important for the initiation and recruitment of chromatin to the transcriptional apparatus [67]. 

### Ovarian Intracellular cAMP and Stimulation of PKA Are Increased upon Gonadotropin Activity

Hormones that use the cAMP second messenger system are involved in maintaining female reproductive function and health. Among them are the hormones playing an important regulatory role, such as hypothalamic releasing hormone, gonadotropin-releasing hormone (GnRH), two anterior pituitary gonadotropins—follicle-stimulating hormone (FSH) and luteinizing hormone (LH), and human chorionic gonadotrophin (hCG) which is normally produced by trophoblast cells [68]. Although LH and hCG share a high degree of homology, they serve different roles in reproduction. LH mostly participates in steroidogenesis and ovulation, while hCG causes the persistence of the corpus luteum during early pregnancy and plays role in fetal development [69,70]. The two pituitary gonadotropins, FSH and LH, manifest their functions during all stages of the menstrual cycle, starting from follicle recruitment, through their growth and maturation, to ovulation of the Graafian follicle [71]. Folliculogenesis begins in fetal life and depends on the interaction of many genes and complex machinery of coordinated expression processes [72,73]. FSH and LH by binding to their membrane receptors activate intracellular signaling cascades, the action of which is adapted to the stage and context, and the intended direction is the development of the ovarian follicles [74]. The binding of gonadotropin to the appropriate receptor triggers one of the best-known cAMP/PKA signaling cascades and its action is associated with steroidogenic as well as proliferative effects. It should be noted that each gonadotropic receptor has its own signaling signature depending on the target cell. Thus, the activated intracellular signals, which are responsive to the presence of either FSH or LH, show great similarity but are not interchangeable [71]. As shown in Figure 1 and Figure 2, initiation of the cAMP-dependent signal transduction pathway occurs through the binding of FSH and LH to their G-protein coupled receptors, FSHR and LHR respectively, which are coupled to Gs proteins [43] and induce signaling pathway based on the interaction with some intracellular effectors [75]. Both receptors belong to the GPHR cluster within the rhodopsin family of G-protein coupled receptors (GPCRs) and are characterized by the presence of a large extracellular domain, seven membrane-imbedded α helices connected by intra- and extracellular loops, and the short intracellular domain completing their structure. The expression levels of FSHR and LHR are not constant and are fully dependent on the phase of the ovarian cycle [41]. When FSH binds to the Gαs protein, the intracellular level of cAMP increases due to the activation of tmAC and the conversion of ATP to cAMP [76,77,78,79]. The opposite effect is caused by cyclic nucleotide phosphodiesterases, especially those present in the human ovary PDE3, PDE4, PDE7, and PDE8 [43]. Nevertheless, initiated by LH and FSH signaling increases intracellular cAMP and activates PKA. The binding of cAMP to the PKA R subunits allows active C subunits to phosphorylate cytosolic and nuclear PKA substrates [80]. 

FSH-stimulated PKA activation is likely to result in the regulation of ovarian granulosa cell (GC) metabolism and proliferation, and the production of sex steroids, while LH-induced cAMP/PKA is essential for GC differentiation and the synthesis of steroids required for ovulation [81]. In immature granulosa cells, FSH signals via activation of phosphoinositide-3-kinase–protein kinase B/Akt (PI3K-PKB/Akt) pathway, which is largely mediated by PKA. Activation of PKB (also known as Akt) is associated with increased proliferation, translation, and activation of target genes. During the early follicular phase of the ovarian cycle, relatively high levels of FSH protect granulosa cells of small immature follicles from apoptosis [22]. The FSH-activated cAMP/PKA signaling induces ovarian steroid production and granulosa cell differentiation. The Gαs/cAMP/protein kinase A (PKA) is not the only transduction mechanism triggered by FSHR. FSH also engages the activation of the ERK1/2, which is essential for modulating both proliferation of granulosa cells and steroidogenesis [82]. The actions of PKA are also confirmed in the nucleus indicating the potential involvement of PKA in chromatin remodeling through post-translational modifications of histone H3 and participation in gene transcription. Translocation of the PKA C subunit into the nucleus leads to CREB phosphorylation, which possibly occurs in response to ERK action and as a result of CREB transcription target genes activity [75,82,83]. In addition, it has been shown that the nuclear effects of PKA mediate mitogenic FSH activity in ovarian GC by inducing histone H3 phosphorylation [84,85]. FSH also induces other signaling pathways including AKT, ERK, and p38 MAPK. Their course is related to the regulation of genes involved in the proliferation of the GC layer. In response to FSHR stimulation, AKT, ERK, and p38 MAPK together with the cAMP/PKA pathway form a complex signaling network to control various cellular functions in the ovary [86,87,88] (Figure 1). 

Rodent studies have shown that the LH signaling cascade activates the HIPPO pathway in the granulosa cells in a PKA-dependent manner, and the cAMP/PKA/CREB acts as a signal intermediary [87,89]. This indicates a strong relationship and dependence of the ovulation process on the HIPPO pathway. It has been shown that GC treated with high doses of LH exhibit a transient increase in the phosphorylation level of LATS1, YAP1 and TAZ kinases. The linkage mechanism between cAMP/PKA/CREB and the HIPPO pathway reveals that PKA-dependent phosphorylation of CREB promotes the secretion of members of the epidermal growth factor (EGF) family through the GC wall, contributing to EGF receptor activation. In consequence, the sustained stimulation of mitogen-activated protein kinases 3 and 1 [MAPK3/1] occurs. Also, the ERK1/2 pathway is activated by the induction of HIPPO-regulated genes (Areg, Pgr, and Ptgs2) which are required for LH-induced signal execution [89].

## 3. The Bioinformatics Prediction in Ovarian Cancer

This section presents current findings related to bioinformatics data analysis in OC research, particularly those associated with the cAMP signaling pathway. 

Using computational methods to identify genes, epigenetic factors and important pathways associated with OC has been widely studied recently. Epigenetic modifications that affect gene expression without changing DNA sequence are considered to be crucial for malignant tumor formation. The imbalance of histone acetylation and methylation, which manipulates gene expression, may be involved in the process of carcinogenesis. On the other hand, the genetic profile of OC patients importantly affects tumor development and progression of the disease [90]. Certain ethnic groups are more vulnerable to cancer, which is partly due to accumulated genetic mutations and inherited defects in the reproductive system. However, the Cancer Genome Atlas for HGSOC (high-grade serous ovarian cancer) shows a fairly low number of mutation patterns typical for this disease. The difficulty in matching a homogeneous type of therapy with a larger group of patients is due to the heterogeneity of tumors in HGSOC. The genomic background is believed to be very complex and characterized by large variations in copy numbers. Therefore, HGSOC requires individualized therapy based on the genomic profile of a tumor to identify altered genes and pathways that enable therapeutic intervention [91]. Research based on microarrays has contributed to the selection of several dozen genes related to ovarian cancer biomarkers [92] as well as those related to chemoresistance [93,94]. These studies aimed to obtain the prognostic value of gene expression signatures, which should directly translate into stratification and improve patient prognosis. Zhou et al. [95] used multiple bioinformatics tools to identify differential expression genes and cell signaling pathways between low-potential tumors and ovarian epithelial cancer. Gene ontological analysis showed that a significant group of up-regulated differentially expressed genes (DEGs) was involved in cellular processes or cell function, including positive regulation of the cAMP metabolic process and positive regulation of cAMP-mediated signaling [95].

In recent years, systems biology approaches have largely contributed to cancer research advancement. Studies based on Weighted Correlation Network Analysis (WGCNA) allowed for the identification of new cancer biomarkers [96,97]. Selection of central genes related to the development of OC revealed an increased expression of a variety of proteins, such as the adenosine A1 receptor (ADORA1 receptor), which is coupled to the G1 protein-coupled receptor (GPCR1). High expression of ADORA1 is associated with worse disease-free survival in OC patients. Activated ADORA1 affects cAMP signaling by inhibiting adenylate cyclase activity and, consequently, decreases intracellular cAMP levels [98]. 

It has also been noticed that driver genes are abnormally expressed in neoplasms in response to epigenetic modifications which may begin with a change in DNA methylation or modification of chromatin that persist with neoplastic cell division [99]. Xie et al. have indicated the importance of cancer-testis antigens (CTA) or cancer-testis (CT) genes in OC cells, considering their inherent heterogenic expression patterns and immunogenicity, as a great opportunity to extend the findings on targeted cancer therapy [100,101]. CTAs, as target antigens in OC cells, seem to be a promising option for targeted clinical management because they are expressed in the neoplastic cells only. Among them, MAGE, CT45, BAGE, BORIS, GAGE-1/2, HOM-TES-85, NY-ESO-1, PRAME, PIWIL, and AKAP3/4 were studied [102,103]. The latter exhibits a strong connection with cAMP signaling as AKAP proteins responsible for PKA anchoring operate through binding to regulatory subunits of PKA [101].

Li et al. analyzed the expression of the CREB1 gene in various types of cancer using different TCGA, and GEPIA databases and compared transcriptional levels of CREB1 in cancers with those in normal cells. Interestingly, they found significantly higher expression of the CREB1 gene in OC as compared with normal ovarian tissue. The expression profile also determined overall survival, that is, the higher the expression level of CREB1 mRNA, the shorter the overall survival [104]. Similar observations were obtained from GENT2 analysis from TCGA RNA sequencing data. It has also been shown that there is a negative correlation between both CREB1 and alpha-fetoprotein 2 (ATF2) in the clinical stage. Further research based on the Reactome analysis showed that CREB1 activity affects genes associated with the immune system functioning, metabolism, development, and energy homeostasis. Identified genes related to CREB1, included TP53, AKT1, AKT2, AKT3, MMP9, BCL2, CCND1 and CCNDMoreover, a positive correlation was shown between CREB1 overexpression and TP53, AKT1, and AKT2, but not for AKT3 [90]. Also, tissue microarray analysis for CREB1 mRNA expression in different ovarian cell lines revealed that mRNA levels in early-stage cells were higher than in late-stage cells. Additionally, a potential mutation site in the domain responsible for heteromerization and transactivation (kinase-induced domain, KID) was demonstrated [105]. Similar results were reported by Liu et al. [106]. In the study based on PASTA analysis, authors identified potential gene-related transcription factors in ovarian cancer indicating CREB, ATF3, and RFX1 [106]. 

As reported by Dimitrova et al., hyperactivation of the cAMP-CREB1 axis is associated with the phenomenon of chemotherapy resistance in OC patients. Thanks to the multi-omics systems biology framework InFlo, it was established that the inhibition of CREB1 phosphorylation sensitizes OC cells to platinum therapy and limits tumor recurrence [17]. The involvement of the cAMP/PKA signaling pathway in OC pathogenesis was also demonstrated in other studies. Analyzes based on EWAS and KEGG datasets have shown that HGSOC tumors with higher global DNA methylation represent greater platinum resistance, which may be related to an increased rate of relapse after taxane/platinum treatment [105]. Calura et al. investigated molecular alterations that regulate tumor aggressiveness at early stages in different histological subtypes of EOC [107]. They reported that among five studied histotypes of OC, the SerHigh subtype exhibits the most aggressive phenotype, and the cAMP-PKA-CREB1 axis is significantly important for histotypic specificity, cellular metabolism, cell cycle regulation, MAPK signaling, and apoptosis [107].

## 4. Signal Transduction Pathways as Potential Therapeutic Targets

The most common subtype of ovarian epithelial neoplasms is the high-grade serous OC, which accounts for over 70% of EOC and is the most lethal subtype. Rare epithelial OC includes low-grade serous, mucinous, endometrioid, clear cell, and transitional cell subtypes [108]. Accumulating evidence suggests that HGSOC originates from epithelial precursor lesions of the fallopian tubes [109]. More specifically, the development of ovary HGSC may be preceded by intraepithelial carcinoma growing in the fallopian tubes, namely serous tubal intraepithelial carcinoma (STIC) [110]. It has been found that neoplastic transformation of the fallopian tube epithelium into STIC and further disease progression are accompanied by abnormal functioning of several signaling pathways. Hypoactivation of estrogen receptor (ER) and hyperactivation of Hedgehog and PI3K/AKT/mTOR signaling cascades are involved in a transition to neoplasia, while the decreased activity of androgen receptor (AR) and Wnt pathways contribute to disease progression [111]. Detailed recognition of pathologic signaling at the molecular level provides the mechanistic background for novel therapeutic approaches (Figure 3). For instance, deregulated activation of protein kinases and phosphatases, e.g., ErbB2, EGFR, MAPK, VEGF, mTOR, have become one of the most widely studied tumor therapeutic targets [51,112,113,114,115]. 

The course of the cAMP/PKA/EPAC/CREB pathway appears to be another promising candidate for investigating signal dysfunction and related therapeutic strategy in ovarian cancer. In healthy cells, the acidic microenvironment promotes the degradation of the extracellular matrix, while in neoplastic cells low pH facilitates invasive growth and metastasis [116]. It has been suggested that sAC, as an intracellular pH and metabolic sensor, may play role in a tumorigenesis, since cellular pH and metabolism strongly correlate to tumor proliferation and metastatic spread [117] Studies have shown that sAC expression is diminished in many solid and hematologic human cancers, including embryonic carcinoma, teratoma, seminoma, yolk sac tumor, some types of leukemia, and others [118]. Preliminary data on animal and cell line models demonstrated that loss of sAC activity enhances cellular transformation in vitro, activates the MAP kinase signaling pathway, and induces tumorigenesis [118,119]. However, to our knowledge, reports regarding sAC involvement in OC development are missing. Nevertheless, studies indicating its role in the carcinogenesis of different organs may serve as an inspiration for further research and bring novel solutions that address the management of OC patients. Since tmAC is the most widely studied type of AC, the majority of findings related to the participation of cAMP signaling in OC growth are due to tmAC/cAMP/PKA activity. cAMP as a secondary messenger promotes proliferation, migration, metabolism, and invasion of cancer cells, and most importantly, it may be a potential suppressor of apoptosis [120,121]. It has also been shown that many upregulated proteins in cancer cells can promote cAMP production [122,123,124]. Using the analysis of transcriptome data, Yue et al. [16] have demonstrated that in patients with BRCA1 deficient ovarian cancer the cAMP pathway is significantly activated, which is associated with increased expression of the ADRB1, β-adrenoreceptor genes. Expression of these genes promotes the production of the cyclic nucleotide and regulates the cAMP pathway. Moreover, cAMP has been reported to abolish the effects of the tumor suppressor protein p53 and counteract the apoptotic process induced by DNA damage. In addition, the positive effect of cAMP on the expression of the secretory antagonist of the classical Wnt signaling pathway, Dickkopf-1 (DKK1), has also been noted. DKK1 inhibits the proliferation of T lymphocytes and NK cells enabling the cancer cells to evade and survive attacks from the host immune system. These data indicate an interesting concept that can be used for future research on potential targeted therapy in patients with BRCA1 mutated ovarian cancer [16] cAMP concentration in platinum-resistant cancer cells: OV81.2-CP10 and Aldhpos CP70 is significantly higher than in non-transformed cell lines. In vitro studies have shown that PKA inhibitor H89 decreases the survival of platinum-resistant ovarian cancer cells under adherent and non-adherent conditions. Impairment of PKA function in platinum-resistant cells results in inhibition of CREB1 phosphorylation at Ser-133, which in turn decreases the survival of platinum-resistant cancer cells and inhibits the G2-M cell cycle. As a consequence, the formation of tumor spheres in the cells is significantly reduced. Interestingly, the combination of cisplatin and H89 exhibit additive cytotoxic effects which correlate with decreased levels of CREB1 phosphorylation and an increase in cleaved caspase-3 levels [17]. Other studies have shown that cAMP plays a significant role in the implementation of cellular processes, mainly through the activation of PKA or EPAC.

Overexpression of various PKA subunits serves as a common marker of phenotypic changes in ovarian cancer cells and refers directly to a much worse course and prognosis of the disease [23]. Studies have shown that PKA plays an important role in cell migration and invasion [125,126,127,128,129]. Its action leads to the enhancement of signal transduction and is associated with the early transition to the cell cycle. Using the SKOV-3 cell model, McKenzie et al. have demonstrated that PKA is activated at the leading edge of migrating epithelial cells of ovarian adenocarcinoma [129]. They evidenced that PKA activity and AKAP-mediated PKA anchoring are necessary for extracellular matrix invasion, while inhibition of anchoring of either RI or RII PKA subunits impairs migration of OC epithelial cells and results from dysregulation of the PKA subunits expression pattern in this type of tumor. Accumulating data suggest a correlation between the advanced, aggressive stage of epithelial ovarian cancer and overexpression of PKA type I regulatory subunits RI [23,130]. In addition, the relationship between the level of AKAP3 mRNA and poor prognosis at the advanced stage of EOC has also been demonstrated [103]. CT antigens, including AKAP4, have been identified as typical for malignant neoplasms [131]. The presence of AKAP4 is crucial for various signaling complexes and subcellular organelles and determines the location of PKA anchoring [132] which may be related to the metastatic transfer of ovarian cancer [132]. As mentioned before, the mobility of ovarian epithelial cancer cells is associated with the distribution of PKA activity. Additionally, a prominent role in the mechanical relationship between cell voltage and extracellular matrix stiffness has also been documented. McKenzie et al. [133] have noted that PKA activity is regulated by mechanical signaling, which is important for cell migration. Mechanical stretching depends on the contractility of cellular actomyosin and inhibited PKA significantly impairs mechanically controlled migration [133]. Subsequent research has shown that cAMP induces integrin-based OVAR3 cell adhesion apart from the PKA signaling pathway, in an EPAC- and small Rap1-GTPase—dependent manner. The effect was confirmed by the activation of Gαs-coupled β2 -adrenergic receptor (AR) on OVAR3 cells in response to isoproterenol use. Treatment with isoproterenol significantly increased adhesion to fibronectin and evoked stimulation of the EPAC-Rap1 pathway even in the presence of a PKA-H89 inhibitor, suggesting that the effect was PKA-independent [134]. The adhesion of cells to fibronectin depends on the cAMP/EPAC/Rap1 integrins. Studies on OVAR3 cells showed increased adhesion mediated by Rap1 fibronectin, activated by 8-Br-cAMP. Both Rap1 and the inhibitor of Rap1, Rap1GAP are the substrates for PKA and therefore can modulate the course of the EPAC-Rap1 pathway in terms of adhesion [134]. In vitro studies on SKOV3 and OVCAR3 cells reveal that EPAC1 may be an attractive therapeutic target in the treatment of ovarian cancer [57]. In the ovary, cAMP directly induces the activity of signaling molecules EPAC/GEF. These proteins (EPAC1/RapGEF3 and EPAC2/RapGEF4) bind cAMP and activate small Rap1/2 GTP-binding proteins by the exchange of GDP to GTP. EPAC-Rap1 possibly plays role in the integrin-mediated cell adhesion through cAMP-PKA [134]. Studies have shown that EPAC activates Rap1 independently of PKA causing the phosphorylation of PKB, whereas the activity of PKA leads to the inhibition of PKB phosphorylation [125]. Rap1 is recognized as being responsible for integrin-mediated cellular functions, acting as a modulator of cytoskeleton dynamics [135]. Activation of Rap1 by its various regulators, such as guanine exchange factors (GEF) or GTPase activating proteins (GAP), may impact the mobility of cancer cells and, consequently, metastases in various cancer models [136,137]. It has been demonstrated that suppression of Rap1B by miR-708 reduces migration/invasion of ovarian cancer cells [138]. In vivo and in vitro experiments have shown that silencing of EPAC1 expression inhibits the proliferation of ovarian cancer cells and initiates cell cycle arrest [57]. These studies indicate Rap1 and EPAC1 involvement in OC cell’s growth and migration, being a good inspiration for future targeted therapy. In addition, EPAC1 participates in metabolic reprogramming, and activation of the EPAC1/Rap1 signaling axis suppresses protein O-GlcNAcylation, which is a known regulatory factor of transcriptional events. Increased glucose uptake and O-GlcNAcylation are considered to be general features of cancer cells that support oncogenic signaling and malignant progression [139]. 

Apart from being pivotal regulatory hormones in folliculogenesis and steroidogenesis, the two gonadotropins (FSH and LH) are recognized risk factors of OEC. Their action contributes to the growth of ovarian surface epithelium (OSE) cells, which is associated with the activation of ERK1/2 and PI3K and is related to the regulation of epidermal growth factor receptor (EGFR) levels. Choi et al. have demonstrated that in immortalized human OSE (IOSE) cells, gonadotropin-mediated effects (ERK1/2 and Akt pathway activation) induce the expression of EGFR [140]. 

Recent advances in elucidating the role of CREB in human ovarian cancer have shown its pleiotropic effects. CREB as a key transcriptional cofactor activates various transcriptional cascades and expression of target genes [138]. Many studies confirmed higher CREB1 expression in human ovarian adenocarcinoma and cancer cell lines [141,142,143]. It plays a significant role in gene regulation, mitochondrial homeostasis, cell migration, proliferation, and apoptosis. Upon PKA activation, its catalytic subunits translocate to the nucleus to phosphorylate CREB [144]. Phosphorylation is essential for CREB-dependent transcription. Interaction through a kinase-induced domain with transcriptional coactivators (e.g., CBP) allows for the formation of a CREB/CBP (CREB-binding protein) complex. This complex initiates CREB-dependent transcription by recruiting transcription machinery at the CRE site and binding its target genes to CRE sites in the promoters [145]. It has been shown that CREB operates on lower-order effectors, including PKA-EPAC, and its activity contributes to ovarian cells’ viability, proliferation, and inhibition of apoptosis [146]. Moreover, it participates in the transcriptional activation of ovarian steroidogenic enzymes (StAR, aromatase) and the release of estradiol and progesterone [18,87,147]. In addition, CREB supports the survival of ovarian epithelial cells and, therefore, is considered a key agent in epithelial ovarian cancer growth [18]. CREB is known as a factor on which many signaling pathways focus, and its transcriptional activity depends on various post-translational modifications. Disruption of this delicate system leads to abnormalities in metabolism and facilitates carcinogenesis [19]. Therefore, CREB could be identified as one of the key targets for anticancer therapy. As mentioned before, CREB promotes the proliferation of various types of ovarian cells. Immunohistochemical analyses show that some of the ovarian tumors, including mucinous adenocarcinoma and serous adenocarcinoma, have significantly higher levels of CREB protein when compared to normal ovarian cells. This observation was confirmed on the SKOV-3 human ovarian cell line. Western blot analyzes showed increased levels of phosphorylated CREB and phosphorylated ATF1 in SKOV-3 cells as compared with normal superficial ovarian epithelial cells [NOSE]. In SKOV-3, reducing the level of CREB (RNAi treatment) had no significant effect on apoptosis, but significantly reduced proliferation [142]. CREB mediates signals essential for maintaining cell viability during early embryonic development and regulates specific gene expression and cellular growth [90]. Overexpression of CREB-1 is associated with excessive proliferation and malignant transformation of ovarian cells demonstrating its protooncogene activity in supporting the initiation, progression, and metastasis of the tumor [145]. As evidenced by genome-wide analyzes of CREB binding sites, it also directly regulates other than early genes (more than 4,000 genes with cAMP-responsive elements in their promoters) [66,145]. In addition, the use of techniques silencing its activity has revealed the promoting effects of CREB on proliferation and inhibition of apoptosis [148]. Similar findings have been reported for other CREB isoforms. CREB5 is significantly overexpressed on the mRNA and protein levels in various types of cancer, playing a key role in regulating cell growth, proliferation, differentiation, and regulation of the cell cycle [149,150]. CREB5 overexpression has also been determined in fresh tissue and epithelial cell lines of ovarian cancer with a positive correlation between high CREB5 expression and tumor advancement stage. Assessment of patient survival rate by Kaplan-Meier analysis showed that patients with high CREB5 expression levels have shorter overall survival, without metastases. In contrast, low levels of CREB5 expression are characterized by longer relapse-free survival [143].

Fight against cancer includes not only management of the primary lesion but also its recurrence and metastatic growth. Of particular importance is understanding the complexity of molecular pathogenesis with an extension to issues in the field of epigenetics [151,152]. Over the past two decades, research has revealed a great deal of information about long non-coding RNAs (lncRNAs) longer than 200 nt and their importance for the progression and metastatic spread in many types of cancer, including ovarian adenocarcinoma [152]. LncRNAs confer a very rich diversity on oncogenesis by performing post-translational and transcriptional functions [152], the effects of which are important for the proliferation, invasion, and migration of cancer cells [149,153]. In ovarian cancer, lncRNA DQ786243 regulates the processes of acetylation, methylation, and chromatin remodeling. Characteristic changes in these molecules have been identified in many studies carried out on various neoplasms [154,155,156,157,158]. In both, ovarian cancer tissues and cell lines, lncRNA DQ786243 is characteristically elevated, and by specific binding to microRNA-506 (miR-506) it affects CREB, causing the development of ovarian cancer. An association has been demonstrated between increased lncRNA expression and poor prognosis for ovarian cancer, primarily the pathological T stage. Knockdown lncRNA DQ786243 in SCOV3 and OVCAR3 cell lines induces G0/G1 cell cycle arrest and cell apoptosis. In addition, significantly reduced expression of CREB1 protein was demonstrated in those cells, thus confirming the importance of lncRNA DQ786243 in relation to the amount of CREB1, but also the ability of both factors to modulate the proliferation of ovarian cancer cells. Loss of the cellular ability to migrate, proliferate and invade [141] was shown among additional benefits. Other studies have shown overexpression of HAS2-AS1 (HAS2 Antisense RNA 1), in SCOVInterestingly, the overexpression of HAS2-AS1 was positively correlated with the presence of CREB. The JASPAR CORE database allowed for the recognition of the CREB binding site with the HAS2-AS1 promoter, which was also confirmed by the chromatin immunoprecipitation (ChIP) assay and the luciferase reporter vector. The effect of this binding is to accelerate the level of transcription, which according to Tong et al. [151], promotes the progression of ovarian cancer.

## 5. Therapy

### 5.1. The Potential Role of PDE Inhibition in Ovarian Carcinoma Cells

Although phosphodiesterase inhibitors are used quite widely in the treatment of various diseases, their usage in ovarian cancer therapy is still at the experimental level on animal models and cell lines. For review purposes, we present the activity of roflumilast (selective phosphodiesterase-4 inhibitor) in a therapeutic context. This drug has already been approved by the US Food and Drug Administration (FDA) for psoriasis and COPD treatment. Gong et al. observed that roflumilast increases intracellular cAMP levels by potentiating PKA/CREB signals, which inhibits tumor growth. This phosphodiesterase 4 inhibitor markedly reduces the proliferation of OVCAR3 and SKOV3 cells and significantly stimulates apoptosis. Its action is manifested by the downstream activation of PKA and CREB phosphorylation, confirming the hypothesis about the function of the cAMP/PKA/CREB signaling in the regulation of cancer growth. In addition, roflumilast increases the expression of mitochondrial ferritin (FtMt) with resulting apoptosis and G0/G1 arrest [15]. Similar effects are observed in OVCAR3 and SKOV3 cells treated with cis-diamminedichloroplatinum (cis-DDP). Moreover, roflumilast appears to enhance cisplatin sensitivity and is capable of reversing DDP resistance. The effects of roflumilast, in terms of viability, on OVCAR3-DDP-resistant and SKOV3-DDP-resistant cells are dose-dependent and time-dependent [159]. Reducing the level of CREB not only inhibits the expression of FtMt but also affects PKA activity and CREB phosphorylation. In vitro studies revealed that inhibition of PKA (using an H89 inhibitor) abolishes the effects of roflumilast on cell viability, apoptosis, and proliferation processes. Similarly, as shown in the cell viability and proliferation analyses on OVCAR3 and SKOV3, silencing of CREB (shRNA) significantly attenuates the effect of the PDE4 inhibitor. Also, CREB silencing blocks the effects of roflumilast on apoptosis and G0/G1 arrest [157]. The above results were confirmed by in vivo experiments carried out on nude mouse models and SKOV3-DDP-R xenograft models [15,159].

### 5.2. AMP-PKA-pCREB, Leading to the Rapid Activity of PARP and DNA Repair

According to “precision and personalized medicine”, the future modern therapy designed for patients with ovarian cancer should target specific molecular features in cancer cells. One of the promising therapeutic perspectives is the use of DNA-dependent nuclear enzymes, such as poly(adenosine diphosphate-ribose) polymerase (PARP) [160]. PARP are the main enzymes activated in the response to DNA damage [161]. Currently, PARP inhibitors (PARPi) are used in first-line, second-line, or maintenance therapy after platinum-based chemotherapy [162,163]. The European Medicines Agency (EMA) and the Food and Drug Administration (FDA) approved the PARP inhibitors: olaparib, rucaparib, and niraparib for the treatment of ovarian cancer [164]. However, as shown by clinical trials, approximately 35% of patients are resistant to PARPi treatment [165]. The molecular mechanisms involved in PARPi resistance are still unclear [164]. Studies have shown that PARP1 is important for tumor progression as it regulates the generation of intracellular reactive oxygen species (ROS), but also DNA damage due to oxidative stress. Thus, PARPi contributes to the growth of cancer, proliferation as well as chemoresistance. The action of PARP1 is related to protein compensation and repair of double-strand breaks (DSBs). BRCA1 and BRCA2 are also affected by this effect [166,167,168] Chemotherapy can lead to significant DNA destruction, and chemoresistant cells are characterized by decreased intracellular cAMP levels [169]. Regarding activation of PARP via DNA methyltransferase inhibitors (DNMTi), which is partially dependent on ROS, Pulliam et al. reported that the increased accumulation of ROS provoked by DNMTi guadecitabine (Guad) use is also associated with PKA activation. cAMP/PKA under oxidative stress triggers rapid PARP activation and is a form of cell response to the toxic effects of chemotherapy [170]. Regardless of the BRCA status, in vivo and in vitro experiments have shown that the additional presence of DNMTi increases the cell response to PARPi. It is believed that oxidative stress induces cAMP-PKA-pCREB, leading to the rapid activity of PARP and DNA repair. The inhibition of PKA by H89 results in a decreased activity of the protein kinase and, consequently, in a decreased activity of PARP. Induction of PKA overexpression increases PARP levels. The use of PARPi talazoparib (Talaz) decreases both PARP and PKA concentrations [170].

## 6. Conclusions

Many years of cancer research have made major advances in the understanding of the biology, molecular changes, and different origins of ovarian cancer, thus creating an opportunity for a personalized therapeutic approach and treatment with targeted drugs. Targeting CREB, an important regulator of tumor initiation, progression, and metastasis, may bring a novel promising therapy for patients with ovarian cancer. However, its therapeutic potential has not been clinically studied so far. Nevertheless, perhaps the extension of knowledge about its cellular effects will soon become a good starting point for future clinical trials bringing a breakthrough in ovarian cancer management.

## Figures and Tables

**Figure 1 cells-11-03835-f001:**
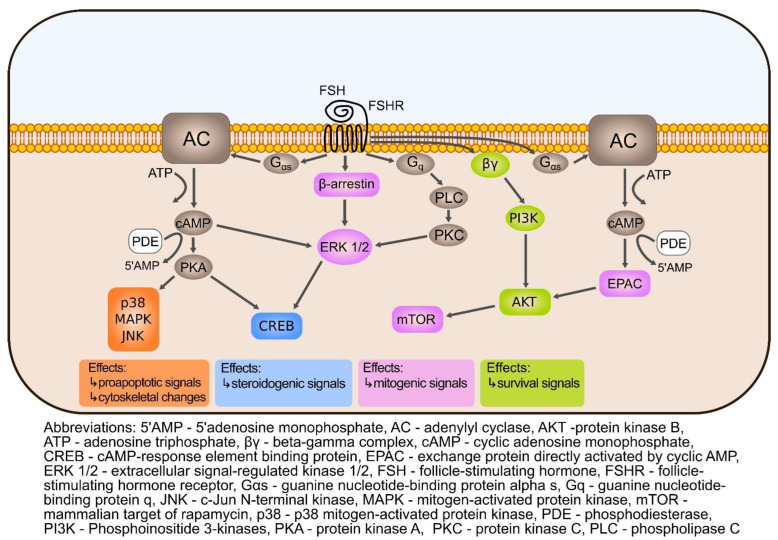
FSH signaling. In response to FSH binding to its specific receptor (FSHR), the corresponding G protein and β-arrestin subunits are activated. Gαs protein-associated signaling stimulates membrane adenylate cyclase (AC) to synthesize cAMP from ATP. PDE opposes the effects of adenylate cyclase by hydrolyzing cAMP. Elevated cAMP activates protein kinase A (PKA), thereby inducing pro-apoptotic signals and cytoskeleton changes (effects related to p38, MAPK, and JNK). cAMP/PKA/CREB activates steroidogenic signals. Epac, as an intracellular cAMP receptor, activates processes related to mitogenic signals (via mTOR) or survival signals (via AKT, the activity of which is also regulated by PI3K). FSH also activates the phospholipase C (PLC) and protein kinase C (PLC/PKC) pathways, which act on ERK1/ERK1/2 may also stimulate CREB.

**Figure 2 cells-11-03835-f002:**
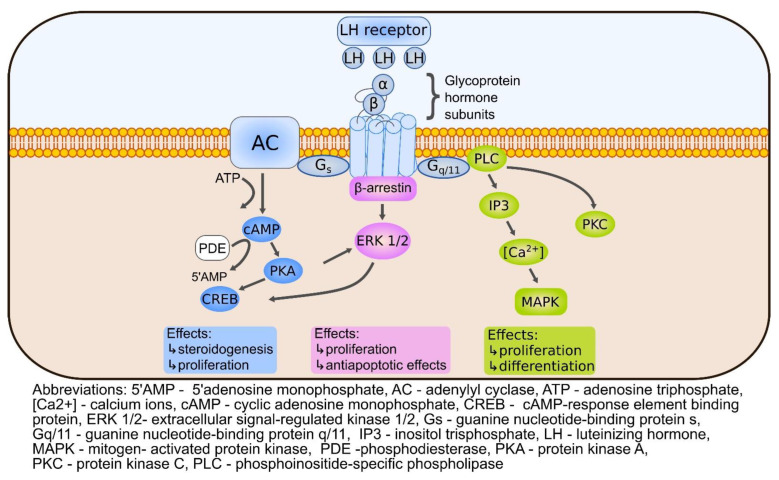
LH signaling. The binding of LH to a specific receptor (LHR) evokes its effects through G proteins (Gs and Gq/11) and β-arrestin. Activated adenylate cyclase (AC) increases the intracellular concentration of cAMP. PDE opposes the effects of adenylate cyclase by hydrolyzing cAMP. cAMP/PKA/CREB activates steroidogenic signals and proliferation. Gq/11 protein induces PLC/IP3/MAPK and/or PLC/PKC pathways stimulating cellular proliferation and differentiation. β-arrestin stimulates ERK1/2, and its action regulates proliferation and inhibits apoptosis.

**Figure 3 cells-11-03835-f003:**
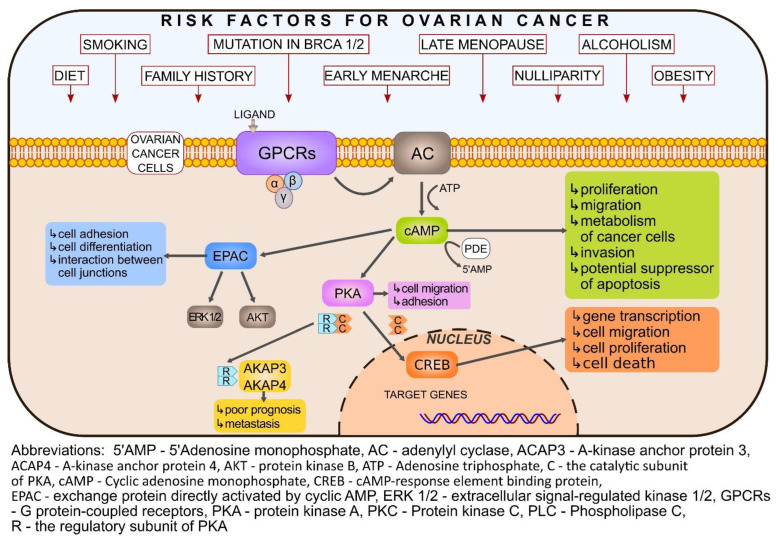
cAMP-dependent pathway in the ovarian cancer cell. Healthy ovarian cell exposed to the risk factors (red boxes) transforms phenotypically into neoplastic cell. Signaling cascades and maintenance of metabolism in the transformed cells may favor all features associated with growth, multiplication, metastasis, and survival. Activation of a cAMP-dependent pathway in the ovary occurs as a result of ligands (e.g., FSH and LH) binding to the G protein-coupled receptor. Changing the conformation of the receptor stimulates adenylate cyclase (AC) and increases cAMP concentration.

## Data Availability

Not applicable.

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
