# Peer review of "cAMP-Dependent Signaling and Ovarian Cancer"

_cells, 2022, doi:10.3390/cells11233835_

Round 1

Reviewer 1 Report

The Review article with the title “cAMP-dependent signalling and ovarian cancer” is an article that contributes to the relative scientific field. However, the manuscript has three major issues. Firstly, the structure of the manuscript does not allow relative comprehension from the reader. I believe that the manuscript should be structured into smaller parts with more relative subtitles and more paragraphs. For example, about the different signaling cascades. Otherwise, the manuscript is extremely confusing, and therefore the aim of a review article is not achieved. Moreover, the use of numerous expressions or words seems unsuitable. For example, the phrases “… primary signaling cascade…” and “… prognosis is negative…”. It is not clear what the authors mean. The Introduction section lacks information about the molecular and signaling background of ovarian cancer. This is a major issue as the background information is not sufficient in order to introduce the topic of the article. Moreover, the first part of section four seems extremely general and does not significantly add information.

Other issues:

·         The Abstract does not properly inform about the contents of this article.

·         It is advised to include more information in the figures in order to achieve better comprehension.

·         The percentage 2.1% refers to the mortality of which total?

·         Figures are not cited in the main text.

·         A conclusion subtitle is missing. Moreover, it is advised to exclude the citation from this part as in the conclusion section no new information should be provided.

Author Response

Dear Reviewer,

The authors would like to thank for comments and constructive remarks, which helped to improve the quality of the manuscript entitled: “cAMP-dependent signaling and ovarian cancer”,

We have taken the comments on a board to improve and clarify the manuscript. Please find below a detailed point-by-point response to all comments (Reviewers’ comments in blue, our replies in black). The revision has been made according to the suggestions of the Reviewers.

Rev. 1

"...the structure of the manuscript does not allow relative comprehension from the reader. I believe that the manuscript should be structured into smaller parts with more relative subtitles and more paragraphs." and “The Abstract does not properly inform about the contents of this article”

Reply: We would like to sincerely thank you for your advice. We understand the Reviewer’s viewpoint here, and we do agree that small sections are more readable. However, because the processes described in this manuscript (especially in the context of ovarian carcinogenesis) are often overlapping and sometimes not clearly described in the literature, it is difficult to present them in a separate parts. Interestingly, since 2010 research on cAMP signalling and its connection with carcinogenesis has been slowed down. This is why some mechanisms are not fully elucidated and might be ambiguous or even confusing. We have revised the manuscript trying to make the text more comprehensible, eg. we’ve rewritten the abstract and introduction sections and made some improvements in the other parts of this paper. We believe that now it seems to be more understandable and that the abstract clearly reflects the content of this manuscript.

the use of numerous expressions or words seems unsuitable. For example, the phrases “… primary signaling cascade…” and “… prognosis is negative…”

Reply: corrected. We’ve replaced “cAMP-dependent pathway is the primary signalling cascade in healthy and neoplastic ovarian cells"  by "cAMP-dependent pathway is one of the most important signalling cascades in healthy and neoplastic ovarian cells". “Negative prognosis” has been replaced by “poor prognosis”.

"The Introduction section lacks information about the molecular and signaling background of ovarian cancer" and "...the first part of section four seems extremely general and does not significantly add information"

Reply: Thank you for your suggestion! As mentioned before, the introduction section has been rewritten and the most important for this review data have been implemented. Indeed, the first part of section four required more detailed description. We’ve shortened this part, changed the title into: “Signal transduction pathways as a potential therapeutic targets”, and added data regarding other than cAMP signalling cascades that are involved in a transition to neoplasia.

“It is advised to include more information in the figures in order to achieve better comprehension”

Reply: Thank you for your opinion. We tried to keep the figures coherent with the main text presenting important steps of cAMP signaling, which are associated with ovarian cancer growth. Additionally, we used a color system, in which different colors refer to the different pathways and their final effects. Figures have titles and short descriptions of these processes, and the main text contains details that help to clarify further consequences of activated pathways.  In our opinion, adding more information to the figures might obscure the purpose of this review and make the graphs confusing.

which are described in detail in the manuscript

“The percentage 2.1% refers to the mortality of which total? “

Reply: Corrected. Indeed, it was not clarified. We have replaced this value by the mortality rate provided by the Global Cancer Statistics in 2020, which is 4.7% and now this sentence is more accurate: “As provided by Global Cancer Statistics in 2020, ovarian cancer (OC) is the 8th most commonly occurring cancer in women with the mortality rate accounting for 4.7% of the entire cancer-related deaths among females

“Figures are not cited in the main text “

Reply: Corrected.

A conclusion subtitle is missing. Moreover, it is advised to exclude the citation from this part

Reply: Corrected.

Reviewer 2 Report

Summary

The review cAMP-dependent signaling and ovarian cancer gives an overview on ovarian cancer followed by a presentation and discussion of pathways and their involvement with the growth of ovarian cancer. This includes specifically PKA that serves as marker of phenotypic changes, and its meaning in diagnosis and prognosis in ovarian cancer, as well as the role of PKA in cell migration and invasion. These facts are related to the signaling pathways involving cAMP-PKA-CREB and consequently possible protein targets for research on therapy are suggested and discussed, respectively the relevant literature is presented.

General comments  

The review is well written and clearly states the relevant arguments that suggest why cAMP dependent processes should be seriously considered as a major determinant in ovarian cancer. It naturally can only give a course overview on ovarian cancer and concentrates on referencing literature relevant to the pathways proposed to be relevant.

The figures/tables/images/schemes are appropriate.

The review does not present much on the initial transformation into cancer and in one place appear to suggest that the cause of the cancer is an (signaling/regulatory) imbalance  (line 413) which I – if that was meant-  would consider controversial. Specifically, any infection aetiology is missing in the overview (see for example DOI: 10.1007/s10096-012-1570-5). In this context, with respect to the discussed Pt-therapy a sentence on HPV would have merit (see PMID: 22331725).

Specific comments

Fig3 and legend:  Though the legend says Activation of the cAMP-dependent pathway occurs as a result of ligand binding to the G protein-coupled receptor” in the figure the ligand is presented merely as “ligand” without further discussion in the text which ligands could or should be expected to stimulate or cause the ovarian cancer. Or does ligand here merely refers to disturbed production of gonadotropins ?

Author Response

Dear Reviewer,

The authors would like to thank for comments and constructive remarks, which helped to improve the quality of the manuscript entitled: “cAMP-dependent signaling and ovarian cancer”,

We have taken the comments on a board to improve and clarify the manuscript. Please find below a detailed point-by-point response to all comments (Reviewers’ comments in blue, our replies in black). The revision has been made according to the suggestions of the Reviewers.

“The review is well written and clearly states the relevant arguments that suggest why cAMP dependent processes should be seriously considered as a major determinant in ovarian cancer. It naturally can only give a course overview on ovarian cancer and concentrates on referencing literature relevant to the pathways proposed to be relevant.

The figures/tables/images/schemes are appropriate.”

Reply: We thank the reviewer for his warm words!

The review does not present much on the initial transformation into cancer and in one place appear to suggest that the cause of the cancer is an (signaling/regulatory) imbalance  (line 413) which I – if that was meant-  would consider controversial.

Specifically, any infection aetiology is missing in the overview (see for example DOI: 10.1007/s10096-012-1570-5). In this context, with respect to the discussed Pt-therapy a sentence on HPV would have merit (see PMID: 22331725).

Reply: We appreciate all your constructive remarks and comments. Chapter four, which now is entitled: “Signal transduction pathways as a potential therapeutic targets”, contains new data regarding cascades that are involved in both: transition to neoplasia and disease progression (doi: 10.1016/j.ygyno.2022.01.027). In our manuscript, we aimed to focus on abnormal functioning of cAMP signalling pathway as a one of the mechanisms that may contribute to cancer growth and as a possible target for therapy.

We mentioned about possible risk factors and infection aethiology in an introduction section and suggested article (DOI: 10.1007/s10096-012-1570-5) has been cited (reference No 5).

“Fig3 and legend:  Though the legend says “Activation of the cAMP-dependent pathway occurs as a result of ligand binding to the G protein-coupled receptor” in the figure the ligand is presented merely as “ligand” without further discussion in the text which ligands could or should be expected to stimulate or cause the ovarian cancer. Or does ligand here merely refers to disturbed production of gonadotropins?”

Reply: Figure 3 presents general stimulation of G protein-coupled receptor. Regarding the subject of this review, we added additional information to the short description under the graph: ” Activation of the cAMP-dependent pathway in the ovary occurs as a result of ligand (eg. FSH and LH) binding to the G protein-coupled receptor”

Stimulation of FSHR and LHR, the two G protein-coupled receptors in the ovary, is described in the main text of this paper.

Reviewer 3 Report

This review summarized the roles of cAMP signaling in ovarian cancer. However, this mansuscript is not well-written. Except for G protein/AC-mediated cAMP synthesis, soluble adenylyl cyclase (sAC; ADCY10) also contributes to the production of cAMP. Bicarbonate and calcium regulate the activity of sAC but not AC (Lines 89-93). In addition, there are many grammatical errors in the manuscript, for example, line 178, line 634, line 663.

Author Response

Dear Reviewer,

The authors would like to thank for comments and constructive remarks, which helped to improve the quality of the manuscript entitled: “cAMP-dependent signaling and ovarian cancer”,

We have taken the comments on a board to improve and clarify the manuscript. Please find below a detailed point-by-point response to all comments (Reviewers’ comments in blue, our replies in black). The revision has been made according to the suggestions of the Reviewers.

“Except for G protein/AC-mediated cAMP synthesis, soluble adenylyl cyclase (sAC; ADCY10) also contributes to the production of cAMP. Bicarbonate and calcium regulate the activity of sAC but not AC”

Reply: We appreciate comments from the reviewer regarding this part of our manuscript. Indeed, bicarbonate, calcium and ATP are the activators of soluble AC. We have corrected this obvious mistake adding a sentence: “Two families of adenylyl cyclase contribute to the production of cAMP, transmembrane adenylyl cyclases (tmACs, ADCY1-9) and, localized within the mitochondria and nucleus, soluble adenylyl cyclase (sAC, ADCY10) the activity of which depends on HCO3- and Ca2+ presence

"…grammatical errors in the manuscript, for example, line 178, line 634, line 663"

Reply: Corrected. The manuscript has been carefully revised in order to correct grammatical errors, including those highlighted by the Reviewer in the last section of the paper.

Section 2.1 is titled now: “Ovarian intracellular cAMP and stimulation of PKA are increased upon gonadotropin activity”.

The sentences:  “The final effect, resulting from PDE4 inhibition, occurs with the full participation of PKA. resulting from PDE4 inhibition, is realized with the full participation of PKA” and “cAMP-PKA-pCREB, leading to the rapid activity of PARP and DNA repair.” have been rewritten with whole section regarding therapy.

Round 2

Reviewer 1 Report

The manuscript has been improved after the first round of revision. I have no new comments to add.

Author Response

Dear Reviewer,
I would like to thank you for your precious time in reviewing our paper and providing valuable comments.
Yours sincerely.

Reviewer 3 Report

This manuscript may be accepted, but need extensive editing of English language.

Author Response

Dear Reviewer,

I would like to thank you for your precious time in reviewing our paper and providing valuable comments.

Yours sincerely.

A. Kilanowska
